# Past and Trends in Cough Sound Acquisition, Automatic Detection and Automatic Classification: A Comparative Review

**DOI:** 10.3390/s22082896

**Published:** 2022-04-10

**Authors:** Antoine Serrurier, Christiane Neuschaefer-Rube, Rainer Röhrig

**Affiliations:** 1Institute of Medical Informatics, University Hospital of the RWTH Aachen, 52057 Aachen, Germany; rroehrig@ukaachen.de; 2Clinic for Phoniatrics, Pedaudiology & Communication Disorders, University Hospital of the RWTH Aachen, 52057 Aachen, Germany; cneuschaefer@ukaachen.de

**Keywords:** cough sound acquisition, automatic cough sound processing, cough diagnosis, cough recognition, literature review, machine learning, quantitative analysis

## Abstract

Cough is a very common symptom and the most frequent reason for seeking medical advice. Optimized care goes inevitably through an adapted recording of this symptom and automatic processing. This study provides an updated exhaustive quantitative review of the field of cough sound acquisition, automatic detection in longer audio sequences and automatic classification of the nature or disease. Related studies were analyzed and metrics extracted and processed to create a quantitative characterization of the state-of-the-art and trends. A list of objective criteria was established to select a subset of the most complete detection studies in the perspective of deployment in clinical practice. One hundred and forty-four studies were short-listed, and a picture of the state-of-the-art technology is drawn. The trend shows an increasing number of classification studies, an increase of the dataset size, in part from crowdsourcing, a rapid increase of COVID-19 studies, the prevalence of smartphones and wearable sensors for the acquisition, and a rapid expansion of deep learning. Finally, a subset of 12 detection studies is identified as the most complete ones. An unequaled quantitative overview is presented. The field shows a remarkable dynamic, boosted by the research on COVID-19 diagnosis, and a perfect adaptation to mobile health.

## 1. Introduction

Cough is a very common symptom, with a prevalence up to 33% of the population, including young children [1]. It is the most frequent reason for which people seek medical advice [1,2,3,4,5,6]. Acute cough can often be a symptom of a common cold clearing itself within two weeks [1,6]. Chronic cough alone counts, however, for 10–38% of requests for respiratory disorders [1,6]. Most cases of chronic cough can be ascribed to smoking or exposure to tobacco smoke [1], as well as to other risk factors such as exposure to environmental pollutants [1,4]. Respiratory and non-respiratory disease conditions of cough include chronic rhinosinusitis, asthma, Chronic Obstructive Pulmonary Disease (COPD), pneumonia, chronic bronchitis, obesity, gastroesophageal reflux disease, lung cancer, heart failure and medications [1,4,7]. These conditions lead to high costs for the health care system, reported at US $40 billion per year for the common cold [6] and US $23 billion per year for asthma and COPD hospitalizations [8] in the United States. The high rate of occurrence and costs call for particular attention. Despite being a very common symptom, the assessment of cough symptoms remains very challenging. A standard assessment approach involves scoring or filling questionnaires [5,6,9,10,11,12], usually performed by the patients themselves or by a parent [12,13], and questioning their objectivity [13,14,15,16]. As a consequence, various studies (see further details in Reference [9], for instance), as well as the ERS Committee [6] “clearly outlined the need for (…) objective cough assessment technology” [17] (p. 2).

Objective approaches were considered as early as the 1960s [6,18,19,20,21]. They relied for a long time on manual assessment (refer, e.g., to Reference [22]), limiting their generalization despite their improved reliability [3,23]. Only recent technological developments since the 2000s, both in the acquisition systems and in technological processing, opened the way for automatic processing [9]. As early as 1937, attempts were made to record and analyze a cough scientifically by measuring diaphragm movements [24]. Since then, several measurements based on different types of sensors have been considered [25]: chest wall movements [26,27,28], airflow measurements [29,30,31,32,33,34], electromyographic measurements [23,35,36], electroglottographic measurements [37], electrocardiographic measurements [27,28], accelerometer-based measurements [38] or a combination of them [17,27,28,35] as a non-exhaustive list. An intuitive, noninvasive and evident marker of coughing is the sound, making it a popular approach for objective cough assessment (see, e.g., Reference [5] for a recent review).

The acquisition and automatic processing of cough sounds constitute the core framework of this review. Its commercial potential can be illustrated by the patents filled in the domain [39,40,41,42,43,44]. Traditionnal acquisition through a microphone are now replaced by advanced techniques based on mobile technologies and wearable sensors. The increasing number of papers trying to diagnose COVID-19 by this way [45,46,47,48,49,50,51,52,53,54,55,56] (see also Reference [57] for a short review) confirm its positive dynamics. Our attention focused more specifically on the acquisition of cough sounds and their automatic detection and classification, i.e., the detection of the cough events in longer audio sequences and the classification of the nature or disease of recorded coughs. These two topics appear often considered together [58] and involve similar technological approaches [59]. Several scientific reviews have already been reported regarding this or a close topic. Chung [3] illustrated the rise of objective cough counter monitors before their automation. Smith et al. [2,19,60] emphasized the need for objective count monitoring and reviewed the latest developments for automatic monitoring at the time. Amoh et al. [8] reviewed the various approaches available for automatic detection. Spinou et al. [9] reviewed the subjective and objective approaches for cough monitoring and focused more specifically on two cough monitoring systems. Pramono et al. [61] provided a systematic review of the detection and classification of adventitious respiratory sounds with a methodology close to the current review. Shi et al. [5], one of the closest and most recent reviews related to the current study, reviewed the existing automatic cough detection systems. Tabatabaei et al. [62] focused on the smartphone-based systems for analyses of respiratory sounds. Recently, the review from Hall et al. [63] focused on the specific tools for counting coughs to measure their frequency. Finally, Lella et al. [57] provided a short review of a COVID-19 diagnosis from the respiratory sounds. Not a review, per se, the ERS task force [6] attempted to provide a normalized scientific framework and to draw up recommendations for the assessment of coughs. All these reviews present limitations regarding our objective. Most of them are outdated, focus on very specific aspects, such as COVID-19 diagnosis or cough counting, or cover side topics. In addition, to our knowledge, no review has covered exhaustively the field of cough detection and classification. The current article intends to fill this gap by providing an updated and complete review of the field. Far more than a simple list of studies, it proposes an unequaled quantitative analysis by (1) providing an up-to-date exhaustive overview, (2) providing quantitative analyses, (3) emphasizing the technological and scientific trends and (4) identifying the most advanced studies towards possible deployment.

The manuscript is organized as follows: the background, in Section 2, describes the coughing process and defines the terms detection and classification, our method is described in the next Section 3, the results of our comparative analyses are in the following Section 4 and Section 5 provides a discussion and conclusion.

## 2. Background

### 2.1. Cough Physiology and Acoustic Properties

Cough is a defense mechanism preventing the entrance in the lower respiratory tract from foreign material and helping to clear the respiratory system from secretions [6,17,64]. Chronic or excessive cough, on the other hand, is the sign of a disorder and requires further investigation [17], the cough itself being only a symptom.

This symptom and some of its properties have already been described about 2000 years ago by physicians such as Aulus Cornelius Celsus and Areteus of Cappadocia (reported in Reference [64]). It is characterized by large inter-subject and intrasubject variability [5,60,65]. A consensus however emerged [19,64], emphasized by the ERS guidelines stating that “all basic scientific articles should refer to cough as a three-phase motor act” [6] (p. 3) (Figure 1): (1) an inspiratory phase, with the glottis fully open, where the air is drawn into the lungs, (2) a compressive phase, with the glottis fully closed, where the respiratory muscles contract and compress the air against the glottis, increasing the pressure, and (3) an expulsive phase, with a sudden reopening of the glottis, where the air is rapidly expelled, the source of the generated cough sound. A subdivision of the expulsive phase, “where the vocal cords briefly close for a second time, producing a further increase in sound at the end of the cough”, can also be considered as a fourth phase [19] (p. 3).

This maneuver generates a cough sound, itself consensually described in three phases [5,19,64,66,67] (Figure 1). The sound phases do not match the motor phases described earlier, leading sometimes to confusion in the literature. These phases are as follows [19]: (1) an explosive phase, short in duration, when the glottis suddenly opens, corresponding to a loud burst, (2) an intermediate phase, lasting longer, corresponding to a noisy steady flow of air, and (3) a voiced phase, when the glottis narrows again and the vocal folds start vibrating. The last phase may not always be present [19,67]. Incidentally, the English word “cough” with the plosive consonant [k] mirrors remarkably these phases [64]. Each of these three acoustic phases are considered to reflect the situation in specific regions of the respiratory system [64,68]: the explosive phase informs about the state of the bronchus, the intermediate phase about the state of the trachea and the voiced phase about the state of the vocal folds and the larynx. This should, however, be balanced with the observation that the role of the “laryngeal structures and the resonance of the nasal and thoracic cavity (…) are uncertain” [5] (p. 2). The oral cavity may also play an important role.

The cough duration is reported around 300–500 ms [63,64,67,69,70] but can also last shorter or longer [65,71,72,73]. The cough definition still remains a challenge, leading to the observation that “a clear definition of cough is lacking in the majority of scientific papers” [6] (p. 3).

For these reasons, there is no universal way of quantifying a cough sound [60]. Monitoring coughs and reporting cough symptoms have, for instance, been performed by counting the number of cough events, of explosive cough sounds or of breaths; by counting the total duration of coughing; by counting the number of epochs (e.g., continuous coughing sounds without a 2-s pause); by measuring the frequency of coughing or by measuring the intensity [6,74]. Automatic processing has used a large variety of features (see the section Features in the Results). Cough sounds are reported to be particularly difficult to differentiate from speech, laughter and throat-clearing [8]. A common strategy consists in detecting its explosive part, considered as more distinctive (e.g., References [27,75]).

### 2.2. Detection vs. Classification

Cough detection, sometimes referred to as cough recognition [76,77,78], refers to the identification and localization of cough events in larger sound files containing many possible audio events, such as noise, speech, TV, laughs, etc. [58]. The purpose is to identify the time range or the instant where a cough occurs. It can be further processed to provide medically relevant information [6]. Cough classification, on the other hand, consists of providing a diagnosis based on a cough sound. It usually assumes that the considered analyzed sound files contain a cough event. It also aims at providing medically relevant information [58,79]. For this task, the considered coughs might sometimes contain extra sound data before and after the event. In this case, a simple coarse detection (as in Reference [17]) is not considered in this review as a pure detector.

## 3. Method

Many variations of the keywords “cough detection”, “cough monitoring” and “cough classification” were searched in Google Scholar between March 2020 and June 2021, leading to a first shortlist of studies. Given the major societal challenge of the COVID-19 pandemic and the relationship with the current reviewed topic, a special search around the word “COVID-19” was performed to capture the latest developments. All studies found related to the reviewed topic were included. Then, the list was iteratively expanded by including all the studies cited in the shortlist and related to the review topic but not already listed so far. The iterative process stopped when no new study could be found. Online archived articles not published elsewhere, whose numbers have strongly increased since 2020, were discarded; one unpublished Ph.D. Thesis was retained [80], and one study was added after the first review [81]. This search and selection process was performed by the first author of the current study.

Google Scholar was not considered as a unique source but as an entry poInternational It generated the first list of relevant studies, incrementally expanded afterwards. Following recent reports [82] recommending the use of Google Scholar in combination with other renowned academic databases, a posteriori searches with PubMed (Medline database), Embase and Web of Science were performed. It led to a slight increase of the listed studies to which the iterative process was applied. Our impression was that the coverage of the field and the number of included studies by this original search strategy appeared much larger than simply relying on the results of database searches. In conclusion, only the studies not referenced or lowly ranked in terms of relevance in the previously mentioned search engines and concomitantly not cited in any of the listed studies would be missing in our review. Each study was individually analyzed by the first author of the current study and transposed in terms of cross-study metrics to seek out quantitative-based evidence and trends. The chosen metrics aimed at covering, at best, all aspects of a study while being general enough to be present in most studies. In the conclusion, we finally draw the accurate profile of a typical study and how this model is evolving. All aspects of a scientific study were covered: (a) the meta-characteristics, including the publication year, the research group, the type of study (detection vs. classification), the classes and the primary motivation, (b) the data, the sensors and acquisition systems and the number and profile of the subjects, (c) the method, including the approach, the exact list of features and the classifiers, (d) the validation, including the cross-validation approach and the numerical results and (e) the number of citations. The metrics were collected from all studies when available by the first author of the current study. All collected metrics were separately processed to draw a picture of the state-of-the-art and of the trends of the domain. The main analyses consisted of interpreting metric distributions, displaying studies and experiments in terms of particular metrics, observing the time evolution of well-chosen metrics and ratios and identifying categories. The results are presented in the next section following the five aspects mentioned above.

A further added value consisted of taking advantage of our large dataset of metadata and metrics to sort out the substantial number of detection studies and select a small subset of the most complete ones. This is presented at the end of the next section.

## 4. Results

### 4.1. Referenced Studies

One hundred and forty-four studies were finally listed. A very large part of them (56%) focused on the detection task and a third (33%) on the classification task. Only 10 studies (7%) considered the two problems: six of them implemented in first place a detector in order to diagnose a disease in the second place [49,54,83,84,85,86], one detected cough into two distinctive classes [87], two alternatively detected coughs and classified coughs according to their type [88] or associated disease [80], and one classified the detected coughs according to the sex of the emitter [78].

#### 4.1.1. Cough Classification Studies

Cough classification consists of labeling a cough event. In most cases, the ultimate objective is to provide a diagnosis for a patient, the subject classification, usually derived from the cough classification results. Both for simplicity reasons and as most of the studies do not specifically address this issue, “cough classification” and “subject classification” will be referred to as “cough classification” in this article. Only a few studies make the distinction (e.g., References [84,89]) and provide explicit separate results (e.g., References [90,91]). The classifications aim at recovering the property of a cough or at diagnosing the presence and/or the type of a pathology. Table 1 provides an exhaustive list of the 58 cough classification studies and their classes.

#### 4.1.2. Cough Detection Studies

Unlike for cough classification, cough detection studies cannot be summarized as in Table 1. Alternatively, the 91 cough detection studies were categorized according to their major motivation, for which five categories were identified (Table 2). This provides an initial overview of the actual underlying motivations for cough detection, even though other decompositions are possible. 

Although all detection studies aim ultimately at maximizing the performance of a detector, some focus more specifically on methodological aspects, possibly at the cost of lower overall performances: the robustness to the data sampling frequency and noise [119,120,121,122], the implementation on a smartphone [123,124,125,126,127], the relevance of the recording sensors [27], the non-intrusiveness of the recording setup [128], the deployment of a system in a real home environment [129], the real-time detection using wearable sensors [130,131], the time variations of the signal [132], the discrimination of individuals [133], the privacy preservation [134,135], the optimization of features [68,136], the tuning of parameters [137], the refinement via a confidence factor [138] and the ability to track a disease [14,139]. Finally, a few studies limit the non-cough events to specific events: speech [140], sleeping movements [123], snoring [74,123,141,142] and sneezing [138].

Technically, cough detection can be considered as a particular instance of classification where the two classes are “cough” and “non-cough”, which justifies considering them together in this review. This also explains why some studies refer to the term “classification” for a detection task (e.g., References [96,143]).

**Table 2 sensors-22-02896-t002:** Cough detection studies per motivation.

Motivation	References	Number
Objective monitoring	[17,22,28,74,75,76,78,80,88,133,140,141,142,143,144,145,146,147,148,149,150,151,152,153,154,155,156,157,158,159,160,161,162,163]	34
Remote/self/lab-free monitoring	[119,120,121,122,123,124,125,126,127,128,129,130,131,134,135,138,164,165,166,167,168,169,170,171,172]	25
Disease assessment	[14,27,69,73,132,137,139,173,174,175,176,177,178,179,180,181,182]	17
Disease diagnosis	[49,54,83,84,85,86,87,183,184,185]	10
Methodology	[68,77,136,186,187]	5

#### 4.1.3. Other Studies

Five studies could not be labeled as classification nor detection but have been partly considered as related to the topic. Four perform automatic data compression or detection of the non-cough segments for further manual counting [188,189,190,191] and another detects respiratory symptoms, cough being only one of them [192]. Conversely, studies of other communities implementing cough detection, such as for body activity detection, usually not driven by medical purposes, were discarded (e.g., References [193,194]).

#### 4.1.4. Publication Years

The distribution of the 144 studies over the publication years is displayed in Figure 2. The research topic started to emerge significantly in 2005 and showed a steady increase until 2020. Earlier research emphasized the need for objective cough processing but usually relied on manual counting [195,196]. In 2005, the needs were clearly identified, and the progress in machine learning made the automatic approach conceivable. The highest number of studies was recorded in 2019 and 2020; the year 2021 may see this number progress even more due to the COVID-19 pandemic. The 144 studies were published by 73 different research groups, corresponding to two publications per research group, on average.

### 4.2. Data and Subjects

#### 4.2.1. Sensors and Acquisition Systems

The recording sensor can be either fixed in the room or directly attached to the subject, a solution particularly adapted for long-term acquisition and monitoring. For the studies concerned with such distinction, 52% of them chose a subject-related acquisition system and 48% a room-related one. In the fix systems, the most frequent solution is to use a microphone or an MP3 recorder ([30,33,58,68,73,80,87,90,91,93,94,97,99,101,104,110,116,128,129,139,149,150,160,161,162,169,172,173,174,178]), or more recently a smartphone as a microphone ([47,48,52,53,54,59,70,74,81,84,85,89,96,102,103,112,113,120,121,122,123,126,135,141,142,171,172]). The subject-attached recording system aims rather at recording during several hours to several days. This is achieved by attaching a microphone or a lapel microphone to the subject ([14,17,22,27,50,88,94,133,136,140,143,145,148,151,152,156,157,158,159,165,166,167,175,176,177,179,180,182,185,188,189,190,191]), by providing a smartphone or a dedicated app to the subject ([78,80,119,120,125,127,134,144,164,192]), by measuring directly the subject internal sounds using a contact microphone ([17,27,75,77,95,100,117,118,146,153,174,181,182,184,188,189,191]) or by considering wearable sensors ([28,29,130,131,138,154,155,168,170,171]). Some studies combine several audio sensors, such as a microphone and a contact microphone [174]. Combining several modalities, such as audio, electromyography and/or chest belt, appears also as a popular solution (e.g., References [17,28]), although the single audio modality seems to outperform multimodal approaches to detect coughs [27]. As displayed in Figure 2, we observe that recordings based on smartphones, wireless and/or wearable sensors tend from the mid-2010s to overtake more traditional systems based on microphones, (see also [62] for a recent review on smartphone-based respiratory sound analysis). The falling trend in the last two years needs however to be confirmed on a long run, the ratio still remaining largely above 1. Not counted in the binary classification of sensors, some studies collect existing material on the web (e.g., Reference [83]). Besides, the crowdsourcing, i.e., the recruitment of volunteers and collection of data from web-based platforms designed on purpose, has seen a recent rapid increase: after a pioneering study carried out over the phone [98], many studies rely on this approach in 2020–2021 for the diagnosis of COVID-19 [45,48,49,51,52,53]. The pandemic, combining major societal and economic impacts and restrictions in the running of in-person experiments, acted as a booster for this technique (see e.g., Reference [47] for a short review). It allows an easy collection of large datasets, at the cost of limited control on the data quality and integrity.

Some studies present full monitoring systems, e.g., embedding acquisition and processing, with a brand name, suggesting that a deployable solution already exists or that a further deployment is in project. They usually consist of ambulatory recording devices aiming for long-term signal recordings, sometimes integrating several sensors. The Hull Automatic Cough Counter (HACC) system [22] includes a lapel microphone, the VitaloJAK system [148,188,189,191] a lapel and a contact microphone, the KarmelSonix system [17] an ambulatory microphone, a contact microphone and non-audio related sensors, the Leicester Cough Monitor (LCM) system [133,176,177] an ambulatory microphone and the LifeShirt system [28] a wearable contact microphone and non-audio related sensors. To our knowledge, only the KarmelSonix system has been approved by the Food and Drug Administration for commercial use, although it has been reported as being not available anymore [63,120]. For mobile-based detection we can report the Automated Device for Asthma Monitoring [106,127,145], the DeepCough [154], the MobiCough [168], the SmartCough [164], the CoughWatch [170,171] (based on smartwatches) and the HealthMode Cough (only the recording component, Reference [144]) systems; for mobile-based classification the TussisWatch [112] and the AI4COVID-19 [54] systems. In addition comes the CoughLoc system [128], based on a network of fix sensors in a room and the SymDetector system [192], a smartphone-based respiratory symptom detector. In this review, only cough audio-based processing is considered. Breath audio and non-audio data-based approaches, such as collected from additional sensors (e.g., in Reference [33]) or from clinical data (e.g., in Reference [58]), have been discarded, unless the data or results from these various sources could not be disentangled (e.g., in References [17,52,107,108,109,114,115]).

#### 4.2.2. Subjects and Protocol

In machine learning, the models are derived from training data and the results evaluated on test data. While a relative flexibility is possible on the training data, the test data are supposed to represent the population targeted by the system. For this reason, we provide in this section a short overview of the overall datasets and a deeper analysis of the test subjects as a marker of the system.

Many studies tend traditionally to rely on a relatively small number of subjects in their database, typically less than 45, emphasizing the difficulty to collect and process such data, as well as the preliminary character of many studies (Figure 3). A few studies have larger databases. The larger ones contained, respectively, 141 [183] and 5320 [45] subjects for the pure detection and classification studies. Two studies considered only one subject [130,131]. As expected, we observed the size of the databases increasing over time, boosted by the recent crowdsourcing data collection approach, as emphasized by the linear regression plotted in Figure 4.

Mirroring the observation made on the overall databases, many studies tend also to rely on a relatively small number of test subjects, typically less than 30 (Figure 5). Testing requires indeed arduous manual labeling of audio data to identify the coughs. The number of samples per subject varied greatly between studies. As a result, the number of subjects characterized the variety of the test data but not their amount. Two studies [45,49] considered, for instance, more than 1000 test subjects. Thirteen other studies tested their systems on more than 100 subjects [29,33,48,52,53,70,81,86,96,107,108,109,114], including all the crowdsourced-based studies for which the number of test subjects was disclosed. All 15 studies dealt logically with cough classification, where the data consisted usually of a few coughs per subject at maximum, unlike cough detection, which usually considered long-term recordings for a few subjects and for which data labeling was much more challenging. The highest number of test subjects for a detection study was reported to be 59 [75]. As noted earlier for the databases, we observed an increase of the number of test subjects in the last years, boosted by the recent research on COVID-19 and crowdsourcing data collection [45,47,48]. Finally, we observed 54% male vs. 46% female test subjects.

The health profiles of the test subjects, according to the categories provided by the authors of the studies, are reported in Figure 6 (The two studies with more than 1000 test subjects [45,49], consisting of 29% COVID-19 patients and 71% healthy subjects, were discarded, as they largely outnumber the number of subjects of the other studies and bias the display). About a third of the test subjects were healthy: for cough detection, they were suitable for a proof-of-concept study, and for cough classification, a pathology is often diagnosed against its absence, necessitating the presence of healthy control subjects. Asthma patients represented about 10% of the test subjects for both study types. COPD, pertussis and bronchitis patients represented each of them about 10% of the test subjects for the detection studies and Lower Respiratory Tract Disease (LRTD) and pneumonia patients again about 10% of the test subjects for the classification studies. Finally, despite discarding two studies [45,49] on the figure, already, 11% of the test subjects of the classification studies were COVID-19 patients, making this disorder the most studied in terms of test subjects. This figure represents the number of patients; representing the number of studies per disease would provide slightly different and complementary information, discarded for space reasons.

Regarding the age category, 85% of the test subjects were adults, and only 15% were adolescents, children, infants or babies grouped into a few studies [30,83,84,85,89,90,91,93,94,102,103,104,105,106,107,108,109,110,111,127,145,150,151,163,190].

Each recording protocol appears unique: controlled environment [33] vs. daily routine [145], clinic [150] or home [128] recordings, day [168] vs. night [73] recordings, resting [160] vs. exercising [17], voluntary [58] vs. spontaneous [121] coughing, one single cough [116] vs. many days [192] for one subject, etc. The data are also processed differently: differentiating between subject and environmental coughs [133] or not [28], considering any kind of real-life background noise [22] or very specific non-cough or background noise [140], considering raw [155] or pre-processed [183] recordings, considering prelabeled [144] or self-annotated data by one or several annotators up to several times [177], etc. Finally, for a given set of data, protocol and method, the choice of the classes plays a major role: one can imagine that discriminating a COVID-19 cough from a pneumonia cough is much more challenging than from a healthy cough. This diversity calls for a careful process while comparing the studies.

### 4.3. Methods

#### 4.3.1. Approach

Both cough classification and detection aim at classifying input data. Interestingly, most of the studies follow the same architecture, divided in four consecutive stages: (1) the preprocessing stage, aiming at preparing the raw signal (resampling, denoising, splitting into successive time frames of 20 to 50 ms, windowing, filtering, etc.), (2) the coarse detection stage, consisting in applying a very simple detection algorithm, usually based on an absolute or adaptive amplitude or energy threshold, to discard the trivial non-cough segments (e.g., References [17,70,88,147,153,164,165,169,192]), (3) the feature extraction stage, consisting in extracting a limited number of relevant features describing the input signal, detailed in the following section Features, and (4) the classification stage, consisting in classifying the input signal on the basis of the features, detailed in the following section Classifiers. In deep learning, becoming slowly the dominant approach, the feature extraction and the classification steps are performed concomitantly by an artificial network, discarding the troublesome task of feature definition [59].

Some studies deviate from this framework. A few detectors can be considered as semi-automatic, or requiring some manual intervention, like the VitaloJAK or the LCM systems. Some studies try also to take advantage of the time variation of the cough signal. This is done by using Hidden Markov Models ([14,68,69,76,133,136,143,145,175,176,177,180]), as already suggested in 1992 [30], by considering time-related neural networks [49,50,56,84,150,155,172], by taking into account time derivatives in the computation of the features ([22,27,68,89,101,128,132,133,136,143,145,148,158,165,166,173,174,175,176,177,180,182]), by considering feature synchronization techniques [132], or by measuring recurrent patterns [51].

#### 4.3.2. Features

An audio signal is a series of values representing a sound at regular time intervals. It provides an extensive description but remains difficult to interpret. It requires further processing, as also performed by the human ear and brain. The aim is to describe the signal in terms of limited representative features.

Many features have been proposed in the literature. The time domain features provide information related to the time evolution of the signal, such as duration and amplitude, whereas the frequency domain features, also called spectral features, provide information related to the frequencies present in the signal. The cepstral domain is obtained by replacing the spectrum by the logarithm of its magnitude and by recalculating back the pseudo-time domain [197]. The cepstral analysis has proven to be very useful in speech and therefore considered as a good candidate for cough analysis.

Up to 178 features have been identified. The features not retained in the final results privileged by the authors have been discarded, as well as the non-audio features (e.g., clinical [29,90] or flow-related features [33]). At the end, the retained number of 178 provides rather an order of magnitude than an exact count. Note also that the dimension of a feature can largely vary, from one single value for the Zero-Crossing Rate (ZCR) [83] to an entire spectrogram image [144] for instance.

The most frequent feature observed is by far the Mel-Frequency Cepstral Coefficients (MFCC), typically a set of 13 coefficients [197], present in 59 studies. It is followed by the ZCR, representing the frequency of zero crossing of the time domain signal and whose values are supposed to be higher for noisier and higher frequency signals, present in 31 studies. The third one is the energy, supposedly high in presence of cough, especially for the explosive phase, present in 14 studies.

We have grouped the features in 17 groups by the type of information they are aiming to capture, each feature being arbitrarily assigned to one group only (Table 3). Although other groupings would also be valid, this provides a general overview. The Figure 7 displays their usage. The cepstral coefficients remain largely the most popular features. This can be attributed to the abundant research in speech recognition [197] largely inspiring the current research. Numerous studies have also tried to optimize them for cough processing ([54,68,158]). Come in second and third positions respectively the time domain features and the measures of power and energy, usually simple features but important markers to discriminate cough, especially the explosive phase. A special type of features are the custom features, sorted in the “Deep learning raw data” category. They correspond to the rising field of deep learning ([29,45,46,47,48,49,50,52,54,55,56,78,80,84,85,86,87,110,126,129,137,143,144,148,149,150,154,155,157,171,172,183,185,186,187]). Deep neural networks take as input large raw data, such as spectrogram images, and provide as output or intermediate results a limited set of machine-interpretable values containing pertinent information, the features. They can then be used for classification in a further step (see the following section Classifiers) [48,183], or the classification stage can be directly embedded within the network, without explicit stage of feature calculation (e.g., References [54,144,154,155]).

In order to keep the number of features minimal, many studies consider a reduction of dimensionality as post-processing. Most of them implement an algorithm selecting the most pertinent subset of features ([27,53,58,70,74,81,83,84,90,91,93,94,96,113,121,122,132,141,142,146,173,174,182,184]) while a few extract the feature principal components ([48,99,134,146]).

#### 4.3.3. Classifiers

27 different classifiers have been identified. The classifiers not retained in the final solution provided by the authors were discarded. The classifiers sharing analogous approaches have been grouped together to form categories. The *Support Vector Machines (SVM) and derivatives category* encompasses the classical SVM classifiers ([48,53,54,74,77,84,89,92,97,101,117,118,121,136,138,158,169,192]), its use in ensemble learning [159] and the Sequential Minimal Optimization, an implementation instance of it ([165,166]). The *Decision Trees and related ensemble learning* category encompasses the classifiers based on a simple Decision Tree ([29,73,88,122,123,130,131,140,151,152,160,161,178,191,192]), as well as the Random Forest classifier, using a multitude of decision trees and taking a final decision by voting ([29,70,74,96,111,112,113,134,135,170,184]), and ensemble decision trees based on boosting ([29,51,59,169]) and bagging [153]. It ranges from very simple trees based on expert observations (i.e., References [73,160,178,192]) to very recent approaches based on ensemble learning (i.e., References [29,153,169]). The *Hidden Markov Models and derivatives* category encompasses the Hidden Markov Models ([69,76,127]), inherited from the speech recognition research and particularly adapted to model times series [176], as well as its implementation where the observation probabilities are generated by Gaussian Mixtures ([14,68,133,145,175,176,177,180]) and by deep neural networks ([143,180]). The *Neural Networks* category encompasses all the neural networks, from the most simple ones with limited size and limited hidden layers, or supposed so ([22,27,69,111,132,156,162,174,182]) to the state-of-the-art deep networks ([29,55,85,87,110,148,149,183]), including the Convolutional Neural Networks ([45,46,47,52,54,78,80,86,126,129,137,144,154,155,171,185,186,187]), the Time Delay Neural Networks ([84,150]), the Recurrent Neural Networks [49,50,56,155,172], the Generative Adversarial Networks [80] and the Octonion Neural Networks [157]. The other classifiers are the *Logistic Regression* ([48,58,75,83,90,91,93,94,95,100,147,163]), *the k-Nearest Neighbors* (kNN) ([49,81,104,119,120,124,125,138,141,142,146,164]), the *Principal Component Analysis* [33], the *Fuzzy C-means* [99], the *Gaussian Mixture Models* ([78,98,102,103,128,168,173]), the *Discriminant Analysis* [30,116] and the *Naïve Bayes* [153]. Their usage is presented in Figure 8: the two most popular classification approaches are the Neural Networks and the Decision Trees and related ensemble learning.

To grasp the trend and popularity of the classifiers, linear regressions of the number of studies on the publication years have been calculated for the last 10 years (2012–2021). Four classifiers or categories show a positive regression line slope, representing their gain in popularity: the Neural Networks (slope = 0.75), the Decision Trees and related ensemble learning (slope = 0.27), the SVM and derivatives (slope = 0.16) and the kNN (slope = 0.22). The popularity of the SVM and derivatives could be attributed to the proven robustness of such classifiers for small datasets, usually used as a reference classifier. The gain in popularity of the kNN could be attributed to its relative simplicity, making it a good candidate for implementation on smartphones with limited capacities, and mostly used lately by the University of the West of Scotland research group ([119,120,124,125,164]). The two remaining categories, the Neural Networks and the Decision Trees and related ensemble learning, present the two highest gains in popularity over the last 10 years, in particular the neural networks, and clearly benefit from the recent breakthroughs in computational powers. These two categories are further detailed in Figure 9: the ensemble learning component and the deep learning component take over the more traditional approaches in their respective categories.

### 4.4. Validation

We have gathered the characteristics of the validation and the numerical results. When several experiments are provided in one study, only the ones considered in the eyes of the authors or by default in our eyes as the most representatives were selected. Although we intend to draw trends, it must be reminded that the studies remain hardly comparable.

The machine learning classifiers usually require a large number of training samples. At the same time, collecting extensive data and manually labeling them remains arduous, limiting the quantity of available data. In addition, a system must be tested on data not used for the training. To optimize the use of available data, several cross-validation approaches are considered. In the *test data* approach, a subset of the data is put aside for validation and not used for training, or another set of data are specifically collected for validation purposes only. In the *Monte**–Carlo* approach, the data are randomly split into a training and a testing subset; this is repeated typically several tens to hundreds of times and the averaged results are provided. While this approach allows a large number of tests, the splitting between training and testing is not controlled. In the *k-Fold* approach, the data are randomly split into k folds of comparable size, the training is done using (k-1) folds and the system tested on the remaining fold; the folds are then rotated so that each fold is used exactly once for testing and the averaged results are provided. Finally, in the *Leave-one-out* approach, the system is trained on all the data except one sample and tested on the left sample; the process is then repeated so that each sample is used once as the test sample. This is a particular case of the k-Fold cross-validation.

We have identified altogether 207 experiments for which the cross-validation process was explicitly mentioned. Half of the experiments consider an explicit set of test data (50%), a quarter a k-Fold cross-validation (27%), followed by a leave-one-out cross-validation (14%); the Monte-Carlo cross-validation is only considered in a minority of experiments (9%). The split between the training and testing data can be considered at the sample (i.e., at the cough event or cough frame) or at the subject level. A subject-level approach can ensure a clear separation between the training and the test subjects. Stricter and more challenging in terms of validation, reported as subject-independent, this approach aims at evaluating the system on unknown subjects, as it is supposed to be the case after the system deployment. A few numbers of systems aim however at implementing a self-training mode where feedback regarding the specific user are possible ([74,133,139,142,164,164,166,176,177]). A majority of the experiments (54%) do not appear however explicitly subject-independent. Only 8% of them consider a self-training mode for which the non-subject-independency could be considered as legitimate.

Reporting results consist in reporting binary classification results, where any input sample is classified as positive or negative. The evaluation is performed on a set of samples for which the classes are already known, usually by mean of human labeling. An actual positive sample classified as positive is a True Positive (TP) and classified as negative is a False Negative (FN). A good system maximizes the number of TP while minimizes the number of FN, measured by the sensitivity ratio TP/(TP + FN), optimally achieving 100%. Complementarily, an actual negative event classified as negative is a True Negative (TN) and classified as positive a False Positive (FP). A good system maximizes the number of TN and minimizes the number of FP, measured by the specificity TN/(TN + FP), optimally achieving 100%. The sensitivity and the specificity measure the performance, as well as a possible bias towards one or the other class. This assumes that the actual number of positive (TP + FN) and negative samples (TN + FP) are of the same order. While this remains usually the case for cough classification, it is not the case for cough detection.

#### 4.4.1. Cough Classification

The number of samples for the two classes are usually of the same order and interchangeable, making the sensitivity and specificity two comparable complementary measures. A sample is usually one cough event. 81 sensitivity-specificity pairs have been gathered, displayed as percentages in Figure 10. Multi-class classifications (e.g., Reference [98]) have been split into several binary classifications and missing values have been recalculated when possible. Most results show comparable values higher than 75% in terms of sensitivity and specificity. The 95% confidence ellipse shows a good balance between sensitivity and specificity, centered around 85%. Further pairwise comparisons between studies remain however very hazardous given their specificities. As an illustration, the number of test subjects ranges from 8 [87] to 1064 [45] and of test cough samples from 16 [79] to 8380 [47]. Some studies consider one sample per subject while some others many samples per subject.

#### 4.4.2. Cough Detection

The cough and non-cough classes are not interchangeable, introducing further challenges. A cough detector typically classes as positive a cough event and as negative any non-cough event: the evaluation still consists in estimating the TP, FN, TN, and FP and ultimately the sensitivity and the specificity.

A first issue is the definition of an event. Some studies simply count the positive and negative processing frames, although they may not be representative of the cough Processing Most studies however aggregate several consecutive frames as events (before or after classification [158]), and count the positive and negative events. A second issue is the definition of a non-cough event: anything else than a cough, with variable durations. One approach consists in considering events of variable lengths, making the processing more challenging, while another consists in dividing the non-cough events into events of similar durations than a cough, at the cost of creating artificial events.

The major issue is that cough segments are usually much more rare than non-cough segments, most of them being trivial to discriminate such as silence. This leads to artificially high values of TN and specificity, not representative of the real performance of the detector: even with a very high specificity, “the number of false positives may exceed the actual cough rate of the individual” [134] (p. 2). This is sometimes solved by discarding trivial non-cough events to keep a similar number of cough and non-cough events, at the cost of losing in authenticity. The alternative is to keep much more non-cough than cough events and aiming for a very high specificity, as emphasized by den Brinker et al. [129], at the cost of losing in readability. A few authors have also decided to replace or complement the count of TN and specificity by a rate of FP per hour. This is also taken into consideration in our review.

We have identified 86 studies having performed 125 experiments with at least one numerical result reported. Further validation measures have been recalculated when possible. The Figure 11 presents the most frequent reported measures: the sensitivity is reported in about 85% of the experiments, while the specificity in less than two-thirds of the experiments (62%). The FP per hour is reported in about 15% of the experiments. The accuracy, representing a global evaluation measure, is reported in a good third of the experiments (38%). Precision, F1 measure and Negative Predictive Value (NPV) represent complementary measures of the classifiers, reported in 15% to 35% of the experiments, as well as the Area Under the Receiver Operating Characteristics (AUC-ROC), which provides an estimation of the power of separability of the two classes. The histograms show a peak of sensitivity around 85–90% and, as expected, a much higher peak of specificity, around 99%. The accuracy, as a global measure, is more equally distributed. The imbalance between the cough and non-cough classes is measured by the ratio of the number of test cough events to the number of test non-cough events, measured at 0.54 ± 0.44 (by excluding the outlier [68] having a much higher number of cough events).

Eighty-seven out of 125 experiments reported a sensitivity-specificity or sensitivity-FP per hour pair (Figure 12). The sensitivity spreads between 70% and 100% and the specificity tends to be higher within a narrower range from 80% to 100%. The pertinence of the FP per hour measure in complement to the specificity can be observed for the study of Sterling et al. [145], who report both high specificity of 96% and high FP per hour of 26.

Finally, the number of cough samples per test subject vary also greatly between studies: it ranges from 43 test cough samples for 84 test subjects [69] to 30,982 test cough samples for 6 test subjects [179]. The median number of test subjects is 13 and of test cough events 764.

### 4.5. Citations

The number of citations measures the impact on the community and highlight the most inspiring studies. They were collected (between October 2020 and June 2021) on Web of Science and Google Scholar. Naturally, older articles are more prone to a higher number of citations. The 10 most-cited studies from each source are displayed (Figure 13), leading to 12 studies altogether. Interestingly, the five most-cited studies are cough detection studies also referenced in Table 4 of the most complete studies (see next section). Three cough classification studies are listed, including the pioneering study of Thorpe et al. [30] and one study from a nearby community ([192]). The most recent study is from 2015.

### 4.6. Most Complete Detection Studies

Unlike classification studies for which small subsets can be easily selected according to their classes (Table 1), the current review identified 91 detection studies hardly manageable. Considering the ultimate objective of detection, i.e., building a system able to detect automatically coughs, our ambition is to rely on the metrics and analyses presented in the previous sections to select a limited subset of studies approaching at best this objective. The selected studies may appear as candidate studies in the perspective of future deployment in clinical environment. For this purpose, a list of objective criteria has been established, covering the main aspects of a study: (1) regarding the study characteristics, it must be published in a peer-review journal, (2) regarding the motivation, it must aim at proposing a complete system and not focusing on a methodological aspect, (3) regarding the data integrity, the samples must not be collected from public web platforms or artificially manipulated, (4) regarding the validation, it must be explicitly subject-independent, unless a self-training mode is implemented, and (5) regarding result reporting, it must report a FP measure (e.g., specificity, false positive rate, or FP per hour). These criteria were designed according to our expertise to target recurrent weaknesses in order to reduce efficiently the number of studies to a manageable size. In no meaning this limited list represents an explicit validation of the integrity of a study. Rather, it constitutes a set of minimum measurable pre-requisites towards this objective. Many crucial parameters have not been considered, such as the number and profile of the test subjects and the recording protocol, as objective criteria could not be set. The manageable size of this subset opens the opportunity for deeper individual reviewing in the future. The studies fulfilling the criteria but not reporting it explicitly could not be selected. This process led to a remaining subset of 12 studies detailed in (Table 4).

## 5. Discussion and Conclusions

One hundred and forty-four studies were shortlisted and split into cough detection and classification, and a comparative review was carried out. It resulted in an overview of the state-of-the-art of the domain and a picture of the trends. Such a quantitative and extensive analyses turned out to be challenging but provided, to the best of our knowledge, an unequalled assessment of the field.

In summary, about half to two-thirds of the studies are dedicated to cough detection and about a third only to cough classification. A rebalancing in favor of more classification studies tends to occur in the last year, pushed by the studies dedicated to COVID-19 diagnosis. Classification studies aim mostly at diagnosing a disorder while detection studies aim mainly at providing an objective or clinical-free cough monitoring or at assessing a disorder. They are mainly applied to healthy subjects and patients suffering from COPD, pertussis, bronchitis, asthma, LRTD, pneumonia and the latest newcomer, COVID-19. A constant rise of publications was observed from 2005 to 2019. A typical study prefers an ambulatory subject-related acquisition system and considers up to 30 test subjects. Smartphones and wearable sensors tend now to overcome traditional acquisition systems. The features are, by far, dominated by the cepstral coefficients, followed by the time domain and the energy-related features. The preferred classifiers are the neural networks, the decision trees and the derived ensemble learning. Among them, deep neural network and ensemble learning show a clear rising trend. About half of the studies adopted a cross-validation scheme based on the rotation of the training and test datasets. Classification studies tend to show comparable sensitivity and specificity performances above 75%, while detection studies show typically sensitivities above 80% and specificities above 95%, due to imbalances between the cough and non-cough classes. A manageable subset of 12 detection studies have been identified as the most complete towards the ultimate objective of a deployable system. As far as we know, only the KarmelSonix system [17] has become a commercial system. All the other studies can be considered as preliminary or in validation phases. Finally, the most influential studies have been identified (including Birring et al. and Barry et al. ([22,177]).

Despite our efforts, some studies may have escaped our attention. We can, however, reasonably speculate that they would not alter the general trends reported in this article.

Despite a thorough analysis, it appeared impossible to compare the efficiency of the methods and rank the studies accordingly. Such a procedure can only make sense for studies sharing similar objectives and data. Alternatively, we attempted to provide an objective overview of the various encountered techniques and how they are evolving. This overview may serve in the future as the basis for selecting limited subsets of studies for which deeper comparative analyses can be reasonably carried out. 

Very close to each other and feeding themselves, cough detection and classification were considered together. While this makes sense when dealing with automatic processing, they remain nonetheless very distinct. To acquire data, cough detection considers arduous long-term monitoring with all the associated challenges, while a smartphone recording a subject for a few seconds in a controlled environment for cough classification is usually enough. Regarding the processing, cough classification aims at classifying two comparable cough signals, usually from balanced classes, while cough detection aims at classifying more rare cough signals against an unlimited number and type of non-cough signals. In other words, cough classification benefit from an easier data acquisition, opening the way for the collection of large datasets, such as collected from crowdsourcing. In cascades, such large datasets open the way for powerful recent deep learning approaches, for which cough classification processing is perfectly adapted. This may explain why cough classification research has become more popular than cough detection research lately and that a rebalancing between the two seems to occur. This phenomenon is perfectly illustrated by the rapidly rising number of COVID-19 classification studies.

Despite abundant research, almost no study has led to a final medical product. This could be attributed to several factors: (1) the difficulty in acquiring data, the fix room-related acquisition systems lacking of flexibility and the ambulatory subject-related ones showing practical limitations; (2) the difficulty in covering all possible real-life situations; (3) the challenge in recruiting subjects and labeling data; (4) the difficulty in covering real-life conditions for the validation and in estimating the false positives and (5) the lack of standardization in the definition of the problem and of the expected outputs, despite existing attempts ([3,6]).

In our view, breakthrough in the field may come from the technology and the methodology. Regarding the technology, the field may embrace two major evolutions: (1) the development of big data and deep learning, boosting the classification performances and (2) the rise of mobile health, making available to every individual at any time an ambulatory audio sensor. The development of big data and deep learning is a very clear trend already observable in the current review. Substantial increases if the dataset sizes have been achieved by means of crowdsourcing lately and used for the diagnosis of COVID-19. Concomitantly, the increase of the dataset sizes opens up the opportunity for deep learning. This is the chosen approach for all recent studies with large datasets. Although the potential of deep learning is unanimously recognized in this domain, it remains arduous to handle: the need for a large set of training data, heavy processing, network architecture and convergence obtained by trials and errors, problems of unbalanced classes, overfitting and generalizability, etc. This still leaves much room for improvement. Mobile-based solutions have now taken the leadership over traditional systems, and optimized approaches are under investigation [121]. An exciting challenge consists in combining big data and deep learning on the one hand and mobile health approaches on the other hand to obtain mobile and robust systems. The current considerations consist of optimizing networks offline and installing the optimized solution on a mobile for a local processing or sending the recorded data to a cloud for a heavier online processing. Regarding the methodology, a standardization of the datasets and expectations may provide a fruitful frame and feed a virtuous competition. Publicly available datasets would furthermore certainly boost research in the domain. This trend is already visible regarding the COVID-19 data collected by means of crowdsourcing [198,199]. Finally, the COVID-19 pandemic, for which cough is one of the most prominent symptoms, and for which a quick, reliable and light diagnosis remains challenging, may emphasize the importance of this line of research and should participate in its promotion.

## Figures and Tables

**Figure 1 sensors-22-02896-f001:**
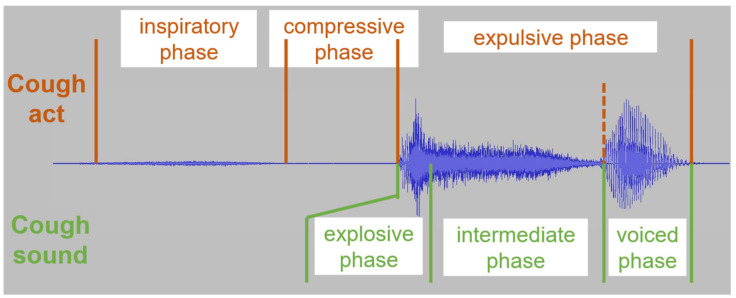
Waveform of a cough superimposed with the phases corresponding to the act (top text, orange) and to the sound (bottom text, green). The horizontal (time) and vertical (amplitude) axes have been omitted to provide a simpler schematic overview.

**Figure 2 sensors-22-02896-f002:**
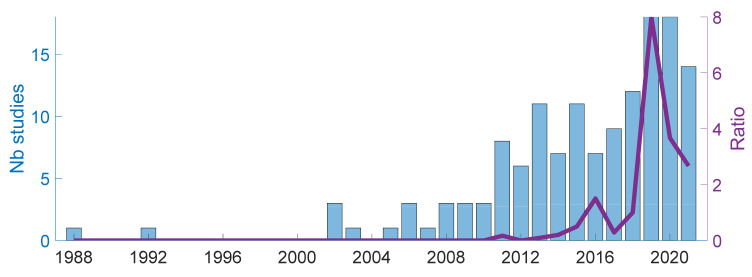
Distribution of the studies over the publication years (blue bars), and ratio of the number of studies using a smartphone or derivative recording system over fix and traditional systems (violet line).

**Figure 3 sensors-22-02896-f003:**
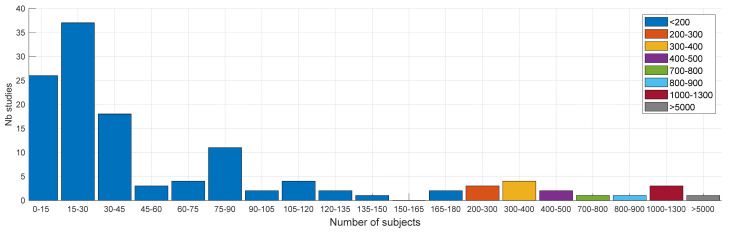
Number of studies vs. overall number of subjects.

**Figure 4 sensors-22-02896-f004:**
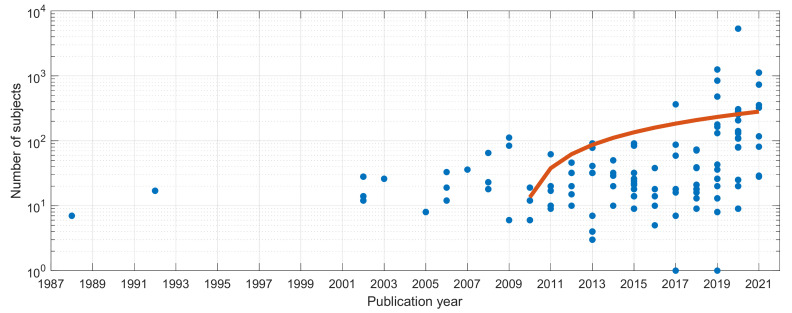
Overall number of subjects vs. publication year per study (blue dots) and associated linear regression (orange line). Note the logarithmic scale of the y-axis.

**Figure 5 sensors-22-02896-f005:**
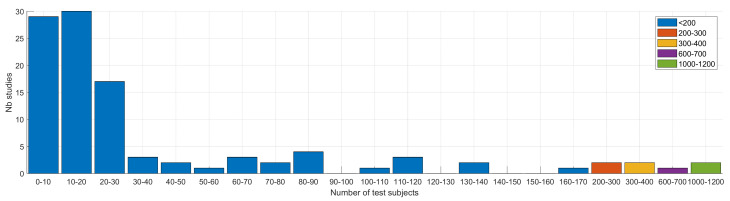
Number of studies vs. number of test subjects.

**Figure 6 sensors-22-02896-f006:**
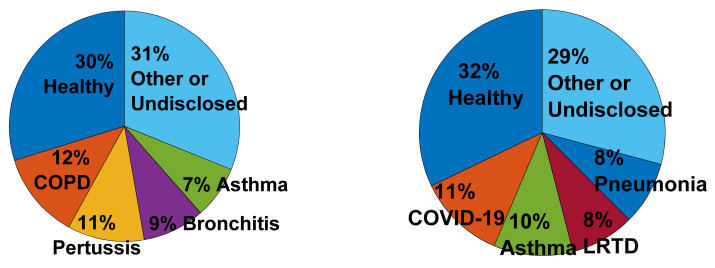
Distribution of the test subjects per health condition, as labeled by the authors, for the detection (**left**) and classification (**right**) studies.

**Figure 7 sensors-22-02896-f007:**
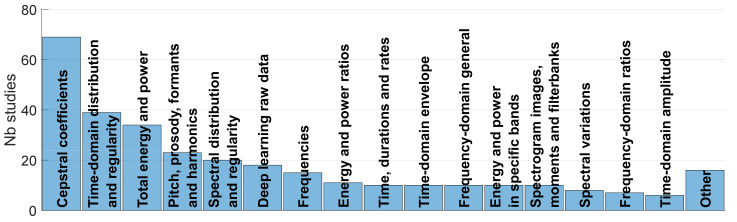
Number of studies for each feature group.

**Figure 8 sensors-22-02896-f008:**
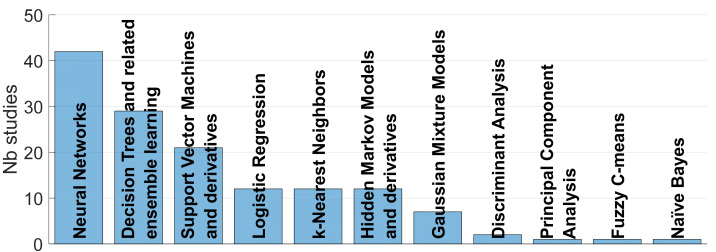
Number of studies for each category of classifiers.

**Figure 9 sensors-22-02896-f009:**
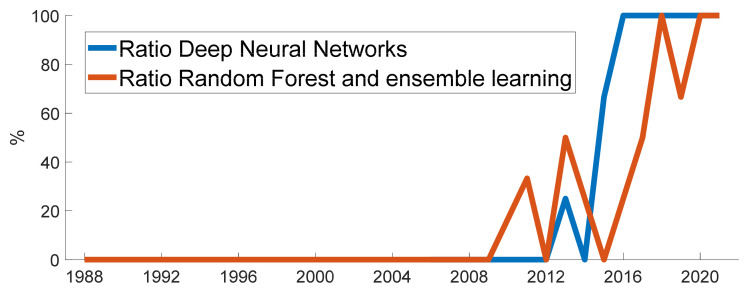
Time evolution of the ratios of the number of deep neural network studies in the “Neural Networks” category (blue) and of the number of random forest and ensemble learning studies in the “Decision Trees and related ensemble learning” category (orange).

**Figure 10 sensors-22-02896-f010:**
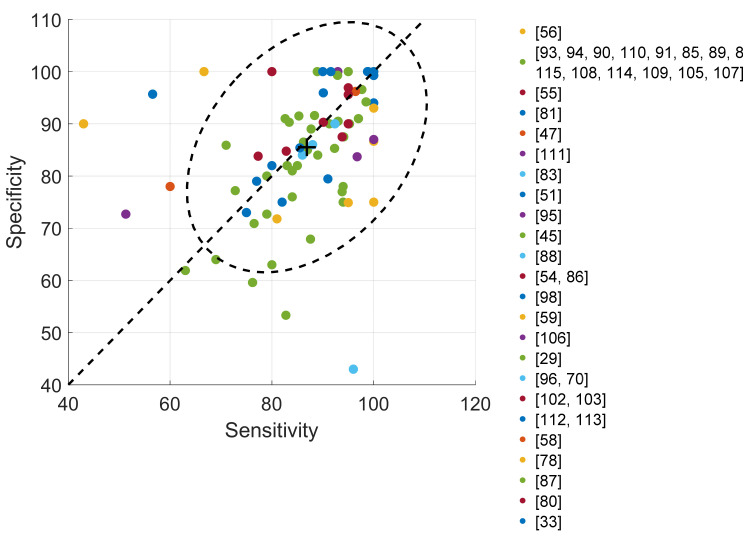
Specificity vs. sensitivity for the experiments of the cough classification studies. Each research group is associated with a specific color. The dashed line represents the specificity = sensitivity line and the dashed ellipse the 95% confidence ellipse.

**Figure 11 sensors-22-02896-f011:**
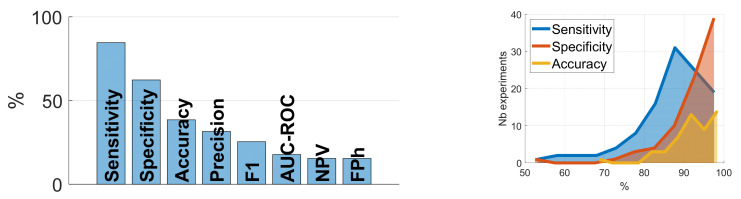
Cough detection. Left: percentage of experiments reporting each of the 8 most frequent measures. Right: histograms of the 3 most-frequent measures.

**Figure 12 sensors-22-02896-f012:**
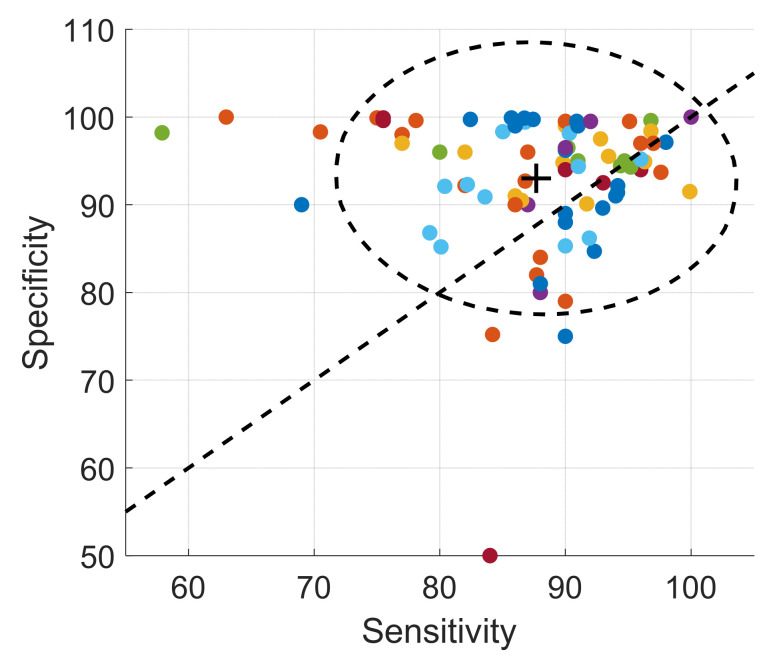
Specificity (top) and FP per hour (bottom) vs. sensitivity for the cough detection. Colors, lines and ellipses are analogous to Figure 10.

**Figure 13 sensors-22-02896-f013:**
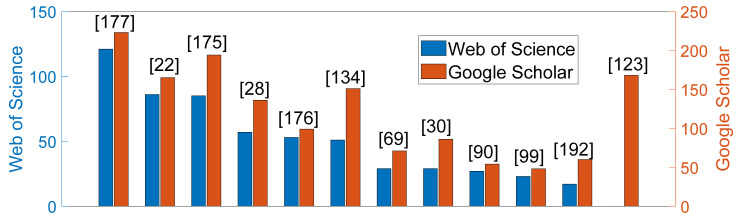
Number of reported citations for the most cited studies.

**Table 1 sensors-22-02896-t001:** Cough classification studies and their classes.

Ref.	Cough Type							Subject Characteristics				Lung Condition			Disease													
	Wet	Dry	Productive	Non-Productive	Spontaneous	Voluntary	Induced	Healthy	Unhealthy	Male	Female	Obstructive	Restrictive	Normal	Croup	Pneumonia	Asthma	Pertussis	COPD	Heart Failure	Tuberculosis	Bronchitis	Bronchiolitis	COVID-19	Cold	LRTD	Upper RespiratoryTract Disease	n/a
[70,79,92,93,94,95,96,97]	X	X																										
[98]		X	X																									
[87]			X	X																								
[99]					X	X																						
[88]						X	X																					
[33,100]								X	X																			
[100]												X		X														
[29]												X	X	X														
[30,101,102,103,104,105,106]								X									X											
[59,85,107]																	X											
[59,84,85,89]															X													
[85,90,91,108]																X												
[105,109]								X								X												
[110]																X	X											
[83,111]																		X										
[112]								X											X	X								
[81]								X												X								
[113,114,115]								X											X									
[80]																					X							
[58]								X													X							
[116]								X									X		X									
[100]																	X		X									
[86]																		X				X	X					
[105]								X															X					
[59]																									X			
[85]																										X		
[105,109]								X																		X		
[85]																											X	
[105]								X																			X	
[85]																							X					
[45,46,47,48,49,50,51,52,53,81]								X																X				
[48,52,54,55]																								X				
[52]																	X							X				
[54]								X										X				X		X				
[56]								X								X		X						X				
[81]								X												X				X				
[78]										X	X																	
[117,118]																												X

**Table 3 sensors-22-02896-t003:** Groups of features.

Group and Description	Examples
**A—Cepstral coefficients**: Coefficients obtained from cepstral analysis.	MFCC [145], improved MFCC [68], Gammatone Frequency Cepstral Coefficients [158]
**B—Time****domain distribution and regularity**: Characterization of the time domain distribution.	ZCR [83], Shannon Entropy [150], Variance [100], Skewness [33], Kurtosis [94], Crest Factor [83]
**C—Total energy and power**: Overall energy or power values, for which the explosive phase is supposed to present a sudden rise.	Total Energy [33], Log-Energy [68], Total Power [156], Average Power [33], Loudness [27]
**D—Pitch, prosody, formants and harmonics**: Speech-related characteristics, supposed to detect voicing activity.	F0 [174], Pitch Standard Deviation [75], Pitch Coverage [75], Formant Frequencies [94]
**E—Spectral distribution and regularity**: Characterization of the spectrum distribution.	Spectral Centroid [29], Spectral Bandwidth [121], Spectral Flatness [112], Skewness [70]
**F—Frequencies**: Peculiar frequencies of the spectrum.	Spectral Rolloff [192], Dominant Frequency [83]
**G—Energy and power in specific bands**: Energy or power values in specific frequency bands.	Power [30], Loudness [27], Log Spectral Energies [58], Octave Analysis [33]
**H—Energy and power ratios**: Ratios of energy or power between different frequency bands.	Power Ratio [79], Relative Energy [132], Relative Power [121]
**I—Time, duration and rates**: Duration and time dynamics values.	Duration [116], Slope [140], L-ratio [33], Left to Right Ratio [192], Rising Envelope Gradient [160]
**J—Frequency domain general**: General spectrum characterization.	Power Spectral Density [99], Spectral Distances [99]
**K—Spectrogram images, moments and filterbanks**: Overall spectrogram images, image moments and outputs of filterbanks.	Local Hu Moments [124], Cochleagram moments [84], Gabor Filterbank [98]
**L—Deep learning raw data**: Input data for deep neural networks.	Mel-Spectrogram [54], Mel-Scaled Filter Banks [137]
**M—Time domain envelope**: Characterization of the time domain envelope shape, capturing typically characteristic peaks in the explosive phase.	Filtered Envelope [69], Peak Number [79], Peak Location [160], Rate of Decay [100]
**N—Frequency domain ratios**: Ratios between different characteristic of the spectrum.	Harmonic to Noise Ratio [132], High-Frequency Content [147], Low quantile ratio [147]
**O—Spectral variations**: Time variation measures of the spectrum.	Spectral Variation [132], Spectral Flux [132], Evo [146]
**P—Time domain amplitude**: Characterization of the time domain amplitude.	Maximum Value [113], Minimum Value [113], Amplitude [160]
**Q—Other**	Wavelet [91], Katz Fractal Dimension [184], DeoxyriboNucleic Acid [81]

**Table 4 sensors-22-02896-t004:** Overview of the selected cough detection studies (see text). GMM-HMM = Gaussian Mixture Model—Hidden Markov Model, TDNN = Time Delay Neural Network, SE = Sensitivity, SP = Specificity, PREC = Precision, FPh = False Positive per hour, ACC = Accuracy and F1 = F1 score.

Study and System	# Test Subjects (Cough Events)	Feature Groups (as in Table 3) + Classifiers	Results	Description
[121]	13 (1309)	A − E − F − H − J − N + SVM	SE = 88–90, SP = 81–75	Robust smartphone-based cough detection
[22] HACC	10 (237)	A + Probabilistic Neural Network	SE = 80, SP = 96	Cough detection over long periods of time for objective monitoring
[14] HACC-LCM	18	A + GMM-HMM	SE = 57.9, SP = 98.2, PREC = 80.9	Objective cough monitoring for COPD patients
[175] LCM	9 (2151/1338)	A + GMM-HMM	SE = 71–82, FPh = 13–7	Continuous cough detection for ambulatory patients
[176] LCM	26/9	A + GMM-HMM	SE = 85.7–90.9, SP = 99.9–99.5, PREC = 94.7, FPh = 0.8–2.5	Continuous cough detection over long periods of time for ambulatory patients
[177] LCM	23/9	A + GMM-HMM	SE = 86–91, SP = 99, FPh = 1–2.5	Continuous cough detection over long periods of time for ambulatory patients
[150]	10/14 (656/1434)	A − B − D + TDNN	SE = 89.8–92.8, SP = 94.8-97.5, ACC = 93.9-97.4	Cough detection for pediatric population
[28] LifeShirt	8 (3645)		SE = 78.1, SP = 99.6, ACC = 99, PREC = 84.6	Cough detection over long periods of time for ambulatory COPD patients
[160]	10 (1019)	I − M − P + Decision Tree	SE = 90.2, SP = 96.5, ACC = 93.1, PREC = 96.7, F1 = 93.3	Cough characterization and detection
[17] KarmelSonix	12		SE = 96–90, SP = 94, PREC = 90–93, FPh = 1.2–1.2	Objective cough monitoring for realistic ambulatory situations
[181]	10 (50)		SE = 84, SP = 50, ACC = 67, PREC = 62.7, F1 = 71.8	Separation of cough and throat clearing sounds
[78]	15 (5489)	A − B − L + CNN	SE = 99.9, SP = 91.5, ACC = 99.8	Overnight smartphone-based cough monitoring for asthma patients

## Data Availability

Not applicable.

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
