# Peer review of "Past and Trends in Cough Sound Acquisition, Automatic Detection and Automatic Classification: A Comparative Review"

_sensors, 2022, doi:10.3390/s22082896_

Round 1
Reviewer 1 Report
This is a thorough review on reported literature on cough signal processing and classification. It is thorough and well organized. It is not novel work on some new approach but it is useful to the community, especially for a new entrant to this field. They authors have done a lot of work and tried to present the whole picture. I have one and single remark and I hope the authors will go the extra mile to do it.
In classification methods the single most important parameter is the dataset. The quality and quantity of the dataset is so important that most of these results are not usually repeatable as we have seen with Covid detection based on Xray data. I have performed a query in your manuscript and the word databases is encountered just once and the word 'database' mostly in the references. What I would strongly suggest is to make the extra effort to include a single Table where the number of the train/test set and the number of individuals is reported in the studies you include. Something like Table 3 and Table 4 you already have just for the data corpus.
To my point of view this is mandatory for 2 reasons:
- Data is more important than features and algorithms because if you have inappropriate data there is nothing the features and models can do
- Many studies report high results but these results are based on an elusive presentation of data. Therefore, they are not repeatable.
There is no need to criticize the studies included but you need to present their database size and number of individual explicitly.
Reviewer 2 Report
It is good design study.
There is a recent study on this subject. The study can be enriched by citing ‘’Automated COVID-19 and Heart Failure Detection Using DNA Pattern Technique with Cough Sounds’’ studies.
Best regards
Reviewer 3 Report
This is an excellent review article on cough classification and detection. The authors did an exhaustive search on the subject, short-listing 143 studies in which 81 were on cough detection, 47 on cough classification, and 10 on both. An exceptionally thorough and methodical analysis then followed. The main contributions are a clear picture of the state-of-the-art, the identified emerging trends, and the high-quality database embedded in the 198 cited references.
This review paper will be a valuable tool for all those entering and working in the field. I would recommend immediate publication in its present form.
Minor typo:
Line 72: should be “objective”
